# Factors Associated with the Delayed Termination of Viral Shedding in COVID-19 Patients with Mild Severity in South Korea

**DOI:** 10.3390/medicina56120659

**Published:** 2020-11-29

**Authors:** Bongyoung Kim, Jang Wook Sohn, Soomin Nam, Jang Won Sohn, Won Suk Choi, Hyoung Seop Kim

**Affiliations:** 1Department of Internal Medicine, College of Medicine, Hanyang University, Seoul 04763, Korea; sobakas@hanyang.ac.kr (B.K.); jwsohn@hanyang.ac.kr (J.W.S.); 2Department of Internal Medicine, College of Medicine, Korea University, Seoul 02841, Korea; jwsohn@korea.ac.kr; 3Department of General Surgery, National Health Insurance Ilsan Hospital, Goyang 10326, Korea; skatnals@nhimc.or.k; 4Department of Physical Medicine and Rehabilitation, National Health Insurance Ilsan Hospital, Goyang 10326, Korea

**Keywords:** COVID-19, coronavirus, extended care facilities, isolation, Republic of Korea

## Abstract

*Background and objectives*: We aimed to analyze factors associated with the period of viral shedding in patients with confirmed COVID-19 who experienced only mild symptoms. *Materials and methods*: We conducted a multicenter retrospective study from three community treatment centers (CTCs) of South Korea. All patients included were admitted to the three centers before 31 March 2020. We collected data about clinical characteristics and the result of real-time reverse transcription polymerase chain reaction (RT-PCR). *Results*: Viral shedding was terminated within 32 days and 36 days in 75% and 90% of patients, respectively (range: 8–49 days). The mean period of viral shedding was 23.8 ± 8.7 days. In the multivariate Cox proportional hazards regression analysis, the existence of underlying comorbidities lowered the probability of the termination of viral shedding (HR = 0.561, 95% CI 0.388–0.812). Female sex and presence of COVID-19-associated symptoms also lowered the probability, but the significance was marginal. *Conclusions*: The existence of underlying comorbidities was associated with delayed termination of viral shedding in COVID-19 patients with mild severity.

## 1. Introduction

At the end of 2019, a novel coronavirus that causes a cluster of pneumonia was found in Wuhan, China [1]. It has been spreading rapidly and has resulted in an epidemic throughout China and the rest of the world. The World Health Organization (WHO) declared a pandemic of Coronavirus Disease 2019 (COVID-19) on 11 March 2020.

The COVID-19 epidemic began earlier in South Korea than in other countries. Since the first confirmed case was found in 19 January 2020 [2], the cumulative number of patients has increased rapidly, notably since the middle of February 2020, after a significant outbreak occurred within a religious group in Daegu and its neighboring areas [3]. As a result of the steep increase in caseloads, the Korean government turned private dormitories and state-run facilities into community treatment centers (CTCs), which provided a place not only for isolation but also for medical support for patients with asymptomatic to mild symptoms. Since the first center was opened on 2 March 2020, a total of 3818 beds from 16 centers distributed across the country were secured for COVID-19 patients with mild symptoms as of 19 March 2020. Indeed, introducing the CTCs was effective with respect to securing hospital beds for severe cases, as well as for prevention of transmission of SARS-CoV-2 within the community.

Despite several advantages, establishing a patient-care process and operating the healthcare system within the centers was challenging due to a lack of evidence about COVID-19 in March 2020. One of the most difficult matters was prospecting the isolation period for patients. Because the result of real-time reverse-transcriptase polymerase chain reaction (RT-PCR) assay was a barometer for the decision of release from isolation at the early phase of pandemic, making the decision of when to perform the follow-up RT-PCR assay without enough evidence was challenging [4]. Furthermore, some patients showed persistent viral shedding longer than 60 days [5]. 

The objective of the present study was to analyze the period of viral shedding for patients with confirmed COVID-19 who experienced mild symptoms. In addition, we analyzed the factors that may affect the period of viral shedding.

## 2. Materials and Methods

### 2.1. Study Setting and Study Population

A multicenter retrospective study was conducted in three CTCs in South Korea. The participating centers were able to admit 136 to 235 COVID-19 patients with mild symptoms, and were located near Daegu, where the largest outbreak occurred in South Korea (Gimje, Gyeongju, and Jecheon). The opening dates of each center were as follows: Gyeongju, 3 March 2020; Jecheon, 9 March 2020; and Gimje, 11 March 2020. As the outbreak was stabilized, some CTCs closed and patients were transferred to other centers: Jecheon closed on 3 April 2020; Gimje, 7 April 2020; and Gyeongju, 14 April 2020. 

All patients were from Daegu and were confirmed to have COVID-19 by RT-PCR assay of nasal and pharyngeal specimens or sputum specimens, and classified as having a mild form of the disease by epidemiologic investigators. According to the classification protocol, patients who met at least one of the following criteria were considered severe and were therefore not admitted to CTC, but were instead immediately hospitalized for treatment: (1) age ≥65 years, (2) presence of underlying comorbidities (e.g., diabetes, chronic kidney disease, chronic liver disease, chronic pulmonary disease, chronic cardiovascular disease, hematologic malignancy, undergoing chemotherapy, and/or use of immunosuppressants, etc.), (3) need for oxygen therapy, and (4) need for special care (e.g., severe obesity, pregnancy, need for renal replacement therapy, etc.) [4]. However, due to the urgent situation in Daegu and unfamiliarity with the novel classification system, some severe cases were admitted to CTCs. Medications for symptomatic treatment, such as antipyretics and antitussives, were available at each center and were prescribed by the medical professionals. Antibiotics or antivirals were not administered to the patients. 

All patients admitted to the three centers up until 31 March 2020 were included in this study; patients who were transferred to hospitals during admission were excluded. 

### 2.2. RT-PCR Test and Discharge Process for Patients

With regard to discharge from the center, an RT-PCR assay of the upper respiratory tract (nasal and pharyngeal specimen) and/or lower respiratory tract specimens (sputum) was used to detect SARS-CoV-2 [6]. RT-PCR was performed at the same laboratory (Green Cross Laboratories, Yongin, Gyeongggi-do, Korea), using Allplex 2019-nCoV assay (Seegene Medical Foundation, Seoul, Korea).

The RT-PCR test and discharge process for the patients is summarized in Figure 1. The test was only performed on patients who did not have a fever without the help of antipyretics and who did not have pulmonary symptoms. The first specimens were collected and the test was performed after 7 days or more from the day of diagnosis. The timing for the subsequent RT-PCR test was determined by the result of the previous test: If the result was negative, another test was performed after a 24 h interval; if the result was positive or inconclusive, another test was performed after 2–7 days, as determined by the center. 

If two series of RT-PCR tests, performed within at least a 24 h interval, displayed negative results for both, patients could be released from the isolation in accordance with the guidelines from the Korea Centers for Disease Control and Prevention [7].

### 2.3. Data Collection

Information on the day of diagnosis of COVID-19 was provided by the Center Accident Investigation Headquarters in South Korea. Basic medical information was collected through a web-based questionnaire or a telephone interview at the time of admission. It consisted of the date of symptom onset, the date of COVID-19 diagnosis, underlying comorbidities, and symptoms associated with COVID-19. The questionnaire for daily health self-monitoring was distributed twice a day and consisted of self-monitored temperatures, symptoms associated with COVID-19, and other healthcare-related questions. The data were collected until the day that each center closed (Jecheon, 3 April 2020; Gimje, 7 April 2020; and Gyeongju, 14 April 2020). 

### 2.4. Definitions

COVID-19-associated symptoms included fever, dyspnea, cough, sputum, nasal congestion, decreased sense of smell or taste, sore throat, and diarrhea. If at least one of these symptoms occurred during the isolation period, we considered the patient to be symptomatic.

The period of viral shedding was defined as the duration from the day of diagnosis to the day when a patient displayed negative results for two series of RT-PCR tests, performed at least at a 24 h interval. If a patient still showed a positive result of RT-PCR test on the day that each center closed, the duration was calculated from the day of diagnosis to the day of the close of the center. It was considered negative when the cycle threshold (Ct) values of all genes were more than 40 cycles. 

### 2.5. Statistical Analysis

The main variables included age, sex, isolation type (solitary or cohort), presence of COVID-19-associated symptoms, and existence of underlying comorbidities. Categorical variables were analyzed using the Chi-square test or Fisher’s extract test. Continuous variables were analyzed through independent t-test or Mann–Whitney U test. Viral shedding period of patients with or without one of the main variables was evaluated by the Kaplan–Meier method using the log-rank test.

Multivariate Cox proportional hazards analysis was performed to identify variables that were associated with the probability of termination of viral shedding. All statistical analyses were carried out using the statistical package, R (version 4.0.0, R Foundation for Statistical Computing). The statistical significance level was set at *p* < 0.05.

### 2.6. Ethics Statements

The study protocol was approved by the Institutional Review Board (IRB) of Korea University Ansan Hospital (IRB number: 2020AS0083, approved on 31 March 2020). The requirement for written informed consent from patients was waived due to the retrospective nature of the study and its impracticability.

## 3. Results

### 3.1. Clinical Characteristics of Patients

A total of 588 patients were admitted to the abovementioned CTCs and 21 patients were transferred to the hospitals. Among them, 20 patients who were initially admitted to the CTC at Cheonan were transferred to the Jecheon center in the middle of their isolation. Finally, a total of 567 patients were enrolled in the present study: 160 (28.2%) from Gimje, 280 (49.4%) from Gyeongju, and 127 (22.4%) from Jecheon (Figure 2). Viral shedding was terminated within 32 days and 36 days in 75% and 90% of patients, respectively (range: 8–49 days) (Figure 3). The mean period of viral shedding was 23.8 ± 8.7 days.

Table 1 shows the patient characteristics. A total of 360 patients (63.5%) were isolated in a solitary room, whereas 207 patients (36.5%) were isolated in the cohort room. There was no significant difference in mean age (36.3 ± 14.1 vs. 34.2 ± 16.2, *p* = 0.119) or in the proportion of females (65.6% vs. 63.8%, *p* = 0.668) between the two groups. The proportion of the patients with underlying comorbidities (6.4% vs. 6.3%, *p* = 0.959) and COVID-19-associated symptoms (78.9% vs. 74.4%, *p* = 0.219) were similar between the two groups. The median number of RT-PCR tests (2.5 vs. 3.0, *p* = 0.084) and the mean period of viral shedding (24.1 ± 8.7 vs. 23.3 ± 8.7, *p* = 0.302) were similar between two groups. The proportion of patients with viral shedding after 24 days were similar as well (46.7% vs. 50.7%, *p* = 0.352). 

There were 129 (22.8%) patients without COVID-19-associated symptoms. The mean age (35.4 ± 14.9 vs. 36.0 ± 15.0, *p* = 0.654) and the proportion of females (63.9% vs. 68.2%, *p* = 0.370) were not significantly different between those with symptoms and those without. The median number of RT-PCR tests (3.0 vs. 3.0, *p* = 0.178) and the mean period of viral shedding (25.0 ± 7.8 vs. 23.4 ± 8.9, *p* = 0.051) were higher in patients with COVID-19-associated symptoms with marginal significance. A higher proportion of patients with COVID-19-associated symptoms shed the virus after 24 days compared to asymptomatic patients (58.9% vs. 45.0%, *p* = 0.005).

### 3.2. Factors Associated with the Probability of Early Termination of Viral Shedding

Kaplan–Meier survival curves show that the existence of underlying comorbidities was statistically significantly related to the probability of early termination of viral shedding (Figure 4). In the multivariate Cox proportional hazards regression analysis, the existence of underlying comorbidities lower the probability of the termination of viral shedding (hazard ratio (HR) = 0.561, 95% confidence interval (CI) 0.388–0.812, *p* = 0.002). Female sex (HR = 0.835, 95% CI 0.693–1.005, *p* = 0.057) and presence of COVID-19-associated symptoms (HR = 0.810, 95% CI 0.652–1.007, *p* = 0.058) also lower the probability, but the significance was marginal (Table 2).

## 4. Discussion

Knowledge about the period of viral shedding in mild COVID-19 patients is important because more than 80% of the COVID-19 cases involve mild symptoms and are unlikely to require hospitalization [8]. 

As no therapeutic agents and vaccines are currently available for COVID-19, infection control to limit transmission is an essential component of counteracting the novel virus, SARS-CoV-2. Because the main mode of transmission is person-to-person spread via respiratory droplets, one of the vital strategies of infection control measures is isolation for suspected or confirmed cases. Although containment strategies have been adopted and are being performed in most parts of the world, there are limited data about the required length of isolation period for confirmed mild cases. According to the WHO, the recovery time of the disease appears to be around 2 to 6 weeks according to its severity [9]. In addition, a multicenter study in China showed that the median length of hospitalization for non-severe cases was 11 days [8]. 

We found that 75% and 90% of patients obtained two negative RT-PCR tests for SARS-CoV-2 on sequential nasopharyngeal specimens collected ≥ 24 h apart within 32 days and 36 days, respectively. Indeed, the period of viral shedding seems to be longer than that reported in previous studies. A study on the viral load of SARS-CoV-2 in upper respiratory specimens showed that most of the patients (*n* = 14) terminated viral shedding by the 18th day from the onset of symptoms [10]. Another study showed that the median duration of the viral shedding in the respiratory samples of mild COVID-19 patients was 14 days (*n* = 21) [11]. In comparison, similar to our result, a study of 56 patients with mild to moderate symptoms indicated that the median duration of viral shedding was 24 days, and the longest was 42 days [12]. The difference in the viral shedding period among studies may come from the definition of positive and negative results of RT-PCR test. Indeed, we set the cut-off value of Ct as 40 in the present study; this was higher than the cut-off value (Ct = 38) of the study that showed the median viral shedding period as 14 days [11]. However, it does not mean that the live infectious virus is actually released [13]. Recently, the guideline about release from isolation has changed: the result of RT-PCR is no longer an essential barometer to make a decision [14].

We found that the factor that most affected the prolonged viral shedding was existence of underlying comorbidities. In addition, the factors associated with viral shedding vary according to study setting. A Chinese study of 410 patients with recovery revealed that coronary heart disease, albumin levels, and the initial time of antiviral treatment were independent factors associated with SARS-CoV-2 viral shedding [15]. Another study of 113 hospitalized patients found that male sex, delayed admission of hospital, and invasive mechanical ventilation were associated with prolonged viral shedding [16]. In addition, older age and chest tightness were independently associated with delayed clearance of SARS-CoV-2 RNA according to a study of 64 hospitalized patients in China [17]. Unlike previous studies that analyzed hospitalized patients, including severe cases, our study only included mild to asymptomatic cases. Hence, the factors might be somewhat different.

Although the presence of COVID-19-associated symptoms was associated with prolonged viral shedding in this study with marginal significance, it may be another meaningful finding of this study. A study of 71 patients showed that the duration of viral shedding was shorter in asymptomatic patients than symptomatic patients [18]. In addition, a previous study about viral kinetics showed that the patients with severe COVID-19 tend to have a high viral load and a long virus-shedding period [19]. Despite this, asymptomatic or pre-symptomatic patients seem to contribute significantly to the rapid spread of COVID-19. According to some modeling studies, undocumented infections were the majority source (80–90%) of the COVID-19 cases [20,21]. 

Another interesting finding of the present study is that the period of viral shedding was not significantly different between patients who were isolated in a solitary room and those who were isolated in a cohort room. Additionally, cohort isolation did not affect the prolonged viral shedding. This is evidence that the likelihood of re-infection of SARS-CoV-2 is low, at least within a short period of time. If re-infection occurred easily within a short period of time, the patients using the cohort room would have taken a longer period of time to meet the release criteria because of re-infection. Furthermore, this also showed that the cohort strategy could be expanded in settings similar to the CTC, especially for family groups involving children, elderly, or disabled people. 

The main strength of our study is the patient population, in that we were able to yield helpful data using a large number of samples. As shown, all patients enrolled in our study had mild symptoms and only a few of them (6.3%, 32/567) had underlying comorbidities. Since the patients did not require special treatment such as oxygen supplementation or fluid infusion, it would have been difficult to collect medical information if CTCs had not been introduced in South Korea.

There are several potential limitations to our study. First, the protocol of patient care was somewhat different from center to center. In fact, since the centers were opened in an emergent situation, there was no standardized protocol. Second, the collection of clinical symptoms and other medical conditions was dependent on web- or application-based questionnaires, and the information may have been exaggerated or underestimated according to the individuals’ characteristics. To compensate for such a limitation, direct communication or telecommunication was carried out for extraordinary cases and for those who did not respond to questionnaires; the response rate was more than 80% in each center. Third, COVID-19-associated symptoms might not be truly caused by COVID-19; in particular, symptoms that occurred during the maintenance of isolation might be caused by other medical conditions. Fourth, quantitative analysis for viral kinetics was not performed in the current study. Finally, only medical costs were considered, and other factors such as facility operating costs, food expenses, and labor loss were not considered in relation to the estimation of proper timing for the initiation of the RT-PCR test. In addition, medical costs vary among countries and therefore, our result is not applicable to other countries. 

## 5. Conclusions

Of patients with mild COVID-19, viral shedding was terminated within 36 days in 90% of patients and the existence of underlying comorbidities was associated with the delayed termination of viral shedding.

## Figures and Tables

**Figure 1 medicina-56-00659-f001:**
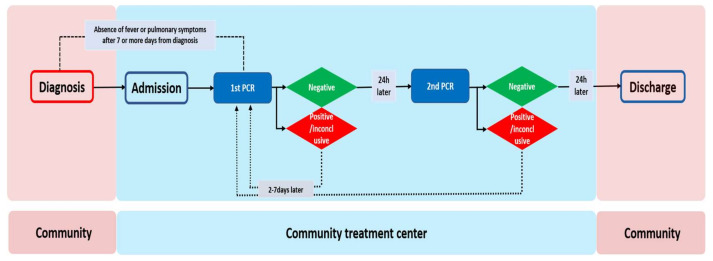
Real-time reverse-transcriptase polymerase chain reaction (RT-PCR) test and discharge process in the Community Treatment Centers.

**Figure 2 medicina-56-00659-f002:**
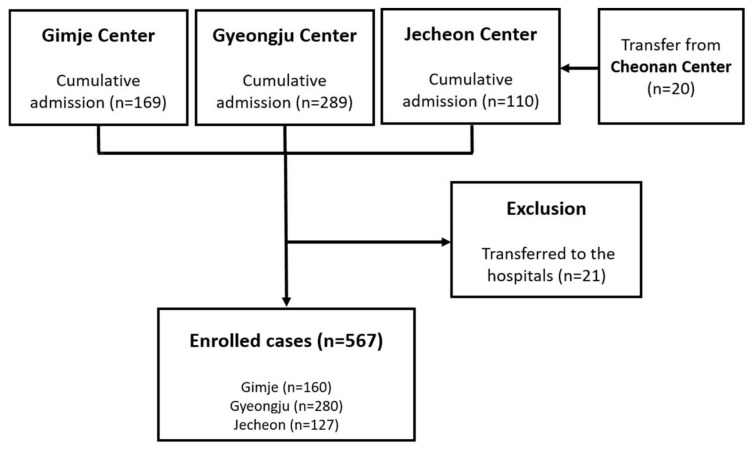
Flow diagram showing the process for selecting cases in this study.

**Figure 3 medicina-56-00659-f003:**
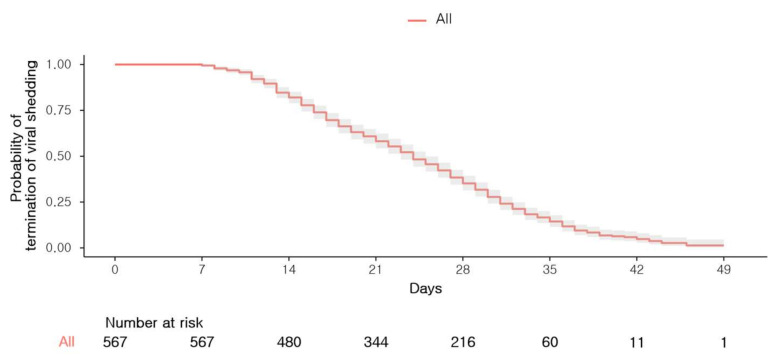
Kaplan–Meier curve for the termination of viral shedding in all patients. The shaded areas represent pointwise 95% confidence intervals.

**Figure 4 medicina-56-00659-f004:**
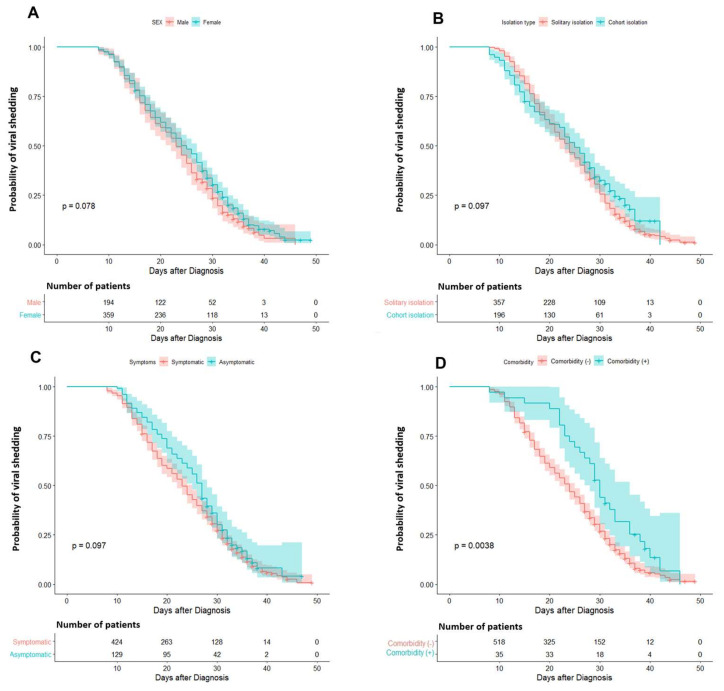
Kaplan–Meier curve for the probability of viral shedding according to the characteristics. (**A**) By sex, (**B**) by isolation type, (**C**) by presence of COVID-19-associated symptoms, and (**D**) by underlying comorbidities. The shaded areas represent pointwise 95% confidence intervals.

**Table 1 medicina-56-00659-t001:** Comparison of clinical characteristics among COVID-19 patients who were admitted to community treatment centers by isolation type and presence of COVID-19-associated symptoms.

		Isolation Type			COVID-19-Associated Symptoms		
	Total (*n* = 567)	Solitary (*n* = 360)	Cohort (*n* = 207)	*p*-Value	Asymptomatic (*n* = 438)	Symptomatic (*n* = 129)	*p*-Value
Centers							
Gimje	160 (28.2)	121 (29.1)	39 (25.8)	<0.001	111 (25.3)	49 (38.0)	0.001
Gyeongju	280 (49.4)	226 (54.3)	54 (35.8)	-	235 (53.7)	45 (34.9)	-
Jecheon	127 (22.4)	13 (3.6)	114 (55.1)		92 (21.0)	35 (27.1)	-
Age, mean ± SD	35.5 ± 14.9	36.3 ± 14.1	34.2 ± 16.2	0.119	35.4 ± 14.9	36.0 ± 15.0	0.654
Female sex (%)	368 (64.9)	236 (65.6)	132 (63.8)	0.668	280 (63.9)	88 (68.2)	0.370
Isolated at the cohort room (%)	207 (36.5)	-	-	-	154 (35.2)	53 (41.1)	0.219
Existence of underlying comorbidities ^a^ (%)	36 (6.3)	23 (6.4)	13 (6.3)	0.959	28 (6.4)	8 (6.2)	0.938
COVID-19-associated symptoms over the course of the disease (%)							
Absence	438 (77.2)	284 (78.9)	154 (74.4)	0.219	-	-	-
Presence	129 (22.8)	76 (21.1)	53 (25.6)		-	-	-
Fever	3 (0.5)	3 (0.8)	0 (0)	0.557	-	-	-
Dyspnea	1 (0.2)	0 (0)	1 (0.5)	0.365	-	-	-
Cough	64 (11.3)	39 (10.8)	25 (12.1)	0.652	-	-	-
Nasal congestion	54 (9.5)	35 (9.7)	19 (9.2)	0.832	-	-	-
Decreased sense of smell or taste	4 (0.7)	2 (0.6)	2 (1.0)	0.625	-	-	-
Others ^b^	52 (9.2)	33 (9.2)	19 (9.2)	0.996	-	-	-
Number of RT-PCR tests, median (IQR)	3.0 (2.0–4.0)	2.5 (2.0–4.0)	3.0 (2.0–4.0)	0.084	3.0 (2.0–4.0)	3.0 (2.0–4.0)	0.178
Period of viral shedding, mean ± SD	23.8 ± 8.7	24.1 ± 8.7	23.3 ± 8.7	0.302	23.4 ± 8.9	25.0 ± 7.8	0.051
Existence of viral shedding after 24 days	273 (48.1)	168 (46.7)	105 (50.7)	0.352	197 (45.0)	76 (58.9)	0.005

Abbreviations: SD, standard deviation; RT-PCR, real-time reverse-transcriptase polymerase chain reaction; IQR, interquartile range ^a^ Includes any chronic disease requiring medication (e.g., diabetes, hypertension) ^b^ Includes sputum, sore throat, or diarrhea.

**Table 2 medicina-56-00659-t002:** Univariate and multivariate Cox regression analysis of variables associated with the probability of early termination of viral shedding.

	Univariate Analysis			Multivariate Analysis		
Variables	Hazard Ratio	95% Confidence Interval	*p*-Value	Hazard Ratio	95% Confidence Interval	*p*-Value
Age	1.000	0.994–1.006	0.965	1.001	0.994–1.007	0.829
Female sex	0.848	0.707–1.018	0.078	0.835	0.693–1.005	0.057
Cohort isolation	0.848	0.701–1.026	0.090	0.868	0.717–1.050	0.146
Presence of COVID-19-associated symptoms ^a^	0.836	0.674–1.036	0.102	0.810	0.652–1.007	0.058
Existence of underlying comorbidities ^b^	0.590	0.410–0.851	0.005	0.561	0.388–0.812	0.002

^a^ Includes fever, dyspnea, cough, nasal congestion, decreased sense of smell or taste, sputum, sore throat, or diarrhea. ^b^ Includes any chronic disease requiring medication (e.g., diabetes, hypertension).

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
