# Peer review of "Factors Associated with the Delayed Termination of Viral Shedding in COVID-19 Patients with Mild Severity in South Korea"

_medicina, 2020, doi:10.3390/medicina56120659_

Round 1

Reviewer 1 Report

Overview:

Thank you very much for the opportunity to review this manuscript. This paper describes retrospectively the viral clearance of COVID-19 patients during isolation and correlates it with external factors like age or underlying diseases. The retrospective analyzation of all available data regarding the current pandemic is of high importance. The paper is well written and well structured. Limitations of the study are due to its retrospective nature. For example, it would have been interesting, to correlate the viral shedding data with antibody response of the patients as well as the determination of viable virus in the samples.

Please find below some minor comments.

Minor comments:

  1. P1, l41: I suggest to rephrase the sentence as follows: e.g. The COVID-19 epidemic began earlier in South Korea… instead of “An epidemic of COVID-19 ….”
  2. P2, l65: 136 to 235 instead of “136-235”
  3. P4, l150-1: “The proportion of patients with persistent viral shedding until the centers were closed were higher among ……” (1) there is some information missing -> when and why were the centers closed? (2) what is the benefit of p-value calculation for this parameter?
  4. Table 1: Please explain the calculation of the p-value in the title in particular to what the p-value was related to, your hypothesis and/or the sense of p-value calculation for the different parameters.

Author Response

Overview:

Thank you very much for the opportunity to review this manuscript. This paper describes retrospectively the viral clearance of COVID-19 patients during isolation and correlates it with external factors like age or underlying diseases. The retrospective analyzation of all available data regarding the current pandemic is of high importance. The paper is well written and well structured. Limitations of the study are due to its retrospective nature. For example, it would have been interesting, to correlate the viral shedding data with antibody response of the patients as well as the determination of viable virus in the samples.

Please find below some minor comments.

Minor comments:

(1) P1, l41: I suggest to rephrase the sentence as follows: e.g. The COVID-19 epidemic began earlier in South Korea… instead of “An epidemic of COVID-19 ….”

-> We changed the sentence following your recommendation.

(2) P2, l65: 136 to 235 instead of “136-235”

-> Done. Thank you.

(3) P4, l150-1: “The proportion of patients with persistent viral shedding until the centers were closed were higher among ……” (1) there is some information missing -> when and why were the centers closed? (2) what is the benefit of p-value calculation for this parameter?

-> The reason and the date for the center closure were added in the method section (line 68-70). We removed the sentence at this round because the clinical meaning is unclear.

(4) Table 1: Please explain the calculation of the p-value in the title in particular to what the p-value was related to, your hypothesis and/or the sense of p-value calculation for the different parameters.

-> We changed the title of Table 1 to “Comparison of clinical characteristics among COVID-19 patients who admitted to community treatment centers by isolation type and presence of COVID-19-associated symptoms”

Reviewer 2 Report

In general, I found this manuscript interesting and well written, describing viral shedding among mild COVID-19 patients and its main determinants. However, time of viral shedding from this work is sligthly longer than that obtained by previous studies. The authors are suggested to discuss more on this point providing exaustive motivations. Notably, a lot of modeling study has been conducted in recent months, and often the infectious period of COVID-19 patients did not reflect what is reported in the current study. Please also discuss on this point considering the following manuscripts: DOI: 10.3390/ijerph17144964; doi: 10.1126/science.abb3221; doi: 10.3390/jcm9051350. Moreover, I would suggest to test for normality all the quantitative variables collected in this study. I suppose that a non-parametric test could be more appropriate for some variables. 

Author Response

In general, I found this manuscript interesting and well written, describing viral shedding among mild COVID-19 patients and its main determinants. However, time of viral shedding from this work is sligthly longer than that obtained by previous studies. The authors are suggested to discuss more on this point providing exaustive motivations.

-> In order to discuss more on the point that the reason for prolonged viral shedding in this study, we added the sentences as follows: “The difference in the viral shedding period among studies may come from the definition of positive and negative results of RT-PCR test. Indeed, we set the cut-off value of Ct as 40 in the present study; which was higher than the cut-off value (Ct = 38) of the study that showed the median viral shedding period as 14 days.” (line 229-232)

Notably, a lot of modeling study has been conducted in recent months, and often the infectious period of COVID-19 patients did not reflect what is reported in the current study. Please also discuss on this point considering the following manuscripts: DOI: 10.3390/ijerph17144964; doi: 10.1126/science.abb3221; doi: 10.3390/jcm9051350. 

-> We added sentences in the discussion reflecting the result of the modeling studies that you mentioned: “Despite it, asymptomatic or pre-symptomatic patients seem to contribute significantly to the rapid spread of COVID-19. According to some modeling studies, undocumented infections were the majority source (80-90%) of the COVID-19 cases [20, 21].” (line 250-252)

Moreover, I would suggest to test for normality all the quantitative variables collected in this study. I suppose that a non-parametric test could be more appropriate for some variables. 

-> Following your recommendation, we tested all factors for normal distribution and found out that ‘Number of RT-PCR tests’ did not have a normal distribution. Mann-whitney U test was performed for 'Number of RT-PCR tests' at this round.

Total

(n=567)

Solitary (n=360)

Solitary (n=360)

Cohort (n=207)

Asymptomatic (n=438)

Symptomatic (n=129)

P-value

Number of RT-PCR tests, median (IQR)

3.0 (2.0-4.0)

2.5 (2.0-4.0)

3.0 (2.0-4.0)

0.084

3.0 (2.0-4.0)

3.0 (2.0-4.0)

0.178

Reviewer 3 Report

The paper is interesting despite the title is not entirely adequate with respect to the conclusions of the study. It would be better to revise the title.

Author Response

The paper is interesting despite the title is not entirely adequate with respect to the conclusions of the study. It would be better to revise the title.

-> We revised the title of the present study as ‘Factors associated with the delayed termination of viral shedding in COVID-19 patients with mild severity in South Korea’. Thank you for the correction.